# Scalable Algorithms for Forest-Based Centrality on Large Graphs

## Abstract

Centrality measures are essential for identifying important nodes and edges within a network. In this paper, we focus on two forest-based centrality measures on undirected graphs: forest node centrality (FNC) and forest edge centrality (FEC), which capture the influence of nodes and edges through their participation in spanning forests. Both centrality measures can be represented using entries of the forest matrix. To address the challenge of computing the two measures on large networks, we propose two scalable algorithms from different perspectives. The first algorithm IFGN combines two variance reduction techniques to approximate the entries of the forest matrix, which is applicable to both FNC and FEC. The second algorithm FECE incorporates a new physical interpretation of FEC, allowing for a better overall estimation. We provide error guarantees for both algorithms and demonstrate their efficiency and effectiveness through extensive experiments on various real-world networks.

## 1 Introduction

Spanning trees and spanning forests of graphs play significant roles in the field of graph data mining. Frequently cited examples include collaborative recommendation [17], opinion dynamics [39, 47], graph density [36], node centrality [24, 43], and others. The forest matrix, denoted as $\Omega = (I + L)^{-1}$ [31, 46], captures important structural properties of graphs, with its entries establishing a connection to the spanning forests of the graph through the forest theorem [8, 12].

Centrality measures based on forests have been proposed and extensively studied [22, 30, 32]. Previous literature has introduced forest closeness centrality [24, 43], which is defined similarly to traditional closeness centrality but utilizes forest distances [11]. It has been demonstrated that forest closeness centrality exhibits superior discriminating power compared to traditional node centrality measures [6] and can be expressed in terms of the diagonal entries of the forest matrix. Specifically, the forest closeness centrality of node $u$ is positively correlated with the reciprocal of the $u^{\text{th}}$ diagonal entry of the forest matrix. Therefore, it can be directly used to represent the importance of node $u$, which is defined as *Forest Node Centrality* (FNC). This measure has been proven to physically represent the average size of the connected component containing node $u$ across all spanning forests rooted at $u$ in the graph [40].

In addition to node centrality, edge centrality serves as a crucial metric in the realm of graph data mining, such as community detection [21], knowledge discovery [37], and visualization of large-scale graphs [15, 29]. Analogous to the definition of FNC, the *Forest Edge Centrality* (FEC) of edge $(u, v)$ is defined as the average size of the connected component containing edge $(u, v)$ across all forests rooted at $u$ [5]. FEC exhibits excellent discriminative capabilities for different edges, and it remains applicable even to disconnected graphs, owing to the fact that a spanning forest can be defined on a disconnected graph. Although enumerating satisfying spanning forests can be time-consuming, it is shown that this metric can be transformed into a combined form of the diagonal and off-diagonal entries of the forest matrix [5].

Such results further underscore the benefits of the forest matrix, indicating that as long as we can compute the entries of the forest matrix, we can calculate FNC and FEC. However, the exact computation of the forest matrix necessitates the calculation of the inverse matrix, which costs $O(n^3)$ operations and $O(n^2)$ memory and thus is impractical for relatively large networks. Therefore, accurately and efficiently estimating the entries of the forest matrix has recently been an important research focus. The mainstream algorithms for approximating the entries of the forest matrix can be categorized into two types, based on the Laplacian Solver and Monte Carlo methods, respectively. Solver-based algorithms, due to limitations in solver performance, do not yield satisfactory approximation results on large-scale networks (exceeding one million nodes). In contrast, Monte Carlo-based sampling algorithms have demonstrated superior performance in experiments and exhibit greater potential.

Based on the forest theorem, the entry in the $u^{\text{th}}$ row and $v^{\text{th}}$ column of the forest matrix can be represented as the proportion of forests in which $u$ rooted at $v$ across all spanning forests of the graph. By incorporating the uniform forest sampling algorithm, research in [40] constructs a simple estimator to approximate each entry. This sampling-based method is applicable to both directed and undirected graphs. Its potential lies in the ability to achieve more precise results by optimizing the sampling process. One optimization approach involves integrating information from neighboring nodes, which has been shown to reduce variance for the estimation of both diagonal [40] and off-diagonal [41] entries and demonstrates improved performance in experiments compared to the original simple algorithm.

Existing methods for calculating forest matrix mainly focus on directed graphs. However, the problem of fast computation of forest matrix on undirected graphs is still of great importance. From the theoretical perspective, FNC and FEC are defined solely on undirected graphs. The matrices corresponding to undirected graphs possess crucial properties, indicating that there is still room for improvement in algorithms applied to directed graphs. From the practical perspective, many downstream graph mining and learning tasks are only defined on undirected graphs. For example, in the task of smoothing graph signals [33], the estimators are proposed on undirected graphs. Additionally, some image analysis tasks are facilitated by undirected random forests [2].

Nevertheless, existing algorithms do not employ effective methods when dealing with undirected graphs. Additionally, for FEC, it is not feasible to directly compute the centrality values for each edge in a short time frame using its definition. Building upon these two observations, we propose algorithms that compute FNC for all nodes and FEC for all edges on undirected graphs. Our main contributions of this paper are summarized as follows.

- We analyze the properties of spanning forests on undirected graphs and devise new estimators for the entries of the forest

matrix. The variance of this estimator is smaller than that of the simple estimator, enabling more precise estimation.

- We combine the strengths of our designed estimator with the neighbor-based estimator[41] in a simple yet non-trivial way, leading to a more accurate and faster estimator. Building on this, we develop the algorithm IFGN.
- Based on the definition of FEC, we identify a new physical interpretation that allows for better computation of FEC. Building on this approach and our variance reduction technique on undirected graphs, we introduce the algorithm FECE.
- We conduct numerical experiments on real-world networks, and the results demonstrate that both IFGN and FECE improve accuracy and speed for the approximation of FNC and FEC.

## 2 Preliminaries

In this section, we introduce some useful notations and fundamental concepts for the convenience of description and analysis.

### 2.1 Graph and Matrix Definitions

Let $\mathcal{G} = (V, E)$ denote an undirected unweighted graph, with node set $V$ and edge set $E \subseteq V \times V$. We define $n = |V|$ and $m = |E|$ as the number of nodes and edges in the graph, respectively. Let $A \in \mathbb{R}^{n \times n}$ be the adjacency matrix of $\mathcal{G}$, with the entry $A_{uv} = 1$ if there exists an edge $e = (u, v)$ from node $u$ to node $v$, and $A_{uv} = 0$ otherwise. For a given node $u$, $N_u$ denote the set of the neighbors of $u$, meaning $N_u = \{v : (u, v) \in E\}$. The degree of node $u$ is $d_u = \sum_{v=1}^{n} A_{uv}$. The degree diagonal matrix of $\mathcal{G}$ is $D = \text{diag}(d_1, d_2, \ldots, d_n)$. The Laplacian matrix $L$ of $\mathcal{G}$ is defined to be $L = D - A$.

### 2.2 Spanning Forest and Forest Matrix

For a graph $\mathcal{G} = (V, E)$, a rooted spanning tree is a connected subgraph of $\mathcal{G}$, where one node is designated as the root. An isolated node is considered as a spanning tree with the root being itself. A spanning forest of $\mathcal{G}$ is a subgraph of $\mathcal{G}$ whose connected components are rooted spanning trees [1, 9].

We use $\mathcal{F}$ to denote the set of all rooted spanning forests. For a given spanning forest $\phi \in \mathcal{F}$, we define a function $r_\phi(u) : V \to V$ mapping node $u$ to the root of its associated connected component. Define $\mathcal{F}_{uv}$ as the set of rooted spanning forests in which nodes $u$ and $v$ are within the same connected component, rooted at node $v$. Formally, $\mathcal{F}_{uv} = \{\phi : r_\phi(u) = v, \phi \in \mathcal{F}\}$. Let $T_\phi(u)$ denote the set of nodes in the connected component containing $u$.

The forest matrix is defined as $\Omega = (I + L)^{-1} = (\omega_{uv})_{n \times n}$. In the context of undirected graphs, the forest matrix $\Omega$ is symmetric and doubly stochastic, meaning that the sum of each row and each column equals 1, with all its entries satisfying $0 \leq \omega_{uv} \leq \omega_{uu} \leq 1$. The entry $\omega_{uv}$ can be encoded by the number of spanning forests, expressed as $\omega_{uv} = |\mathcal{F}_{uv}| / |\mathcal{F}|$ [8, 12].

### 2.3 Forest Node Centrality

Node centrality can be quantified through measures related to the spanning forests of the graph. Previous literature has introduced the concept of forest closeness centrality [24, 43], which is defined similarly to traditional closeness centrality but through forest distances. Specifically, forest closeness centrality is defined as the reciprocal of the average forest distance from a node $u \in V$ to all other nodes

in the graph $\mathcal{G}$. This measure indicates that a node is more central when it has shorter average distances to other nodes.

**DEFINITION 2.1 ([11, 24]).** *For a graph $\mathcal{G} = (V, E)$ and its corresponding forest matrix $\Omega = (\omega_{uv})_{n \times n}$, the forest distance between pair of nodes $u$ and $v$ is defined as: $\rho(u, v) = \omega_{uu} + \omega_{vv} - 2\omega_{uv}$. The forest closeness centrality $C(u)$ for node $u$ is defined as: $C(u) = \frac{n}{\sum_{v \in V \setminus \{u\}} \rho(u,v)} = \frac{n}{n\omega_{uu} + \text{Tr}(\Omega) - 2}$.*

We observe that, in the context of forest closeness centrality, the only term that varies across different nodes $u$ is $\omega_{uu}$, which represents the $u^{\text{th}}$ diagonal entry of the forest matrix. Given that the rest of the expression is constant for all nodes, we can simplify the centrality measure by directly using

$$\text{FNC}(u) = \frac{1}{\omega_{uu}} \qquad (1)$$

as the Forest Node Centrality (FNC) of node $u$. It is easy to verify that using this simplified expression to rank nodes yields the same results as using the original closeness centrality for ranking. In fact, the quantity we defined carries physical significance. According to [40], the reciprocal of $\omega_{uu}$ is equal to the average size of the connected component containing node $u$ over all spanning forests rooted at $u$, expressed as $\frac{1}{\omega_{uu}} = \frac{\sum_{\phi \in \mathcal{F}_{uu}} |T_\phi(u)|}{|\mathcal{F}_{uu}|}$. Intuitively, a node belonging to a larger connected component across all its spanning forests is generally more central and essential.

### 2.4 Forest Edge Centrality

Forest Edge Centrality (FEC) quantifies the centrality of an edge $(u, v)$ by considering its average contribution across all spanning forests $\mathcal{F}_{uv}$ of the graph [5]. The formal definition of FEC is provided in Definition 2.2.

**DEFINITION 2.2.** *Let $\mathcal{G} = (V, E)$ be an undirected, unweighted graph. The FEC of edge $(u, v)$ is defined as the average size of the trees containing $(u, v)$ over all spanning forests in $\mathcal{F}_{vu}$, namely*

$$\text{FEC}(u, v) = \frac{1}{|\mathcal{F}_{vu}|} \sum_{\phi \in \mathcal{F}_{vu}, (u,v) \in \phi} |T_\phi(u)|. \qquad (2)$$

Definition 2.2 indicates that $\text{FEC}(u, v)$ is equal to the expected number of nodes in the tree including edge $(u, v)$, which is part of a spanning rooted forest chosen randomly from $\mathcal{F}_{vu}$. Since enumerating all spanning forests is time-consuming, Lemma 2.3 proposes a new formulation to calculate FEC by expressing the FEC of every edge in terms of the entries for the forest matrix $\Omega$.

**LEMMA 2.3 ([5]).** *Let $\mathcal{G} = (V, E)$ be an undirected unweighted graph with forest matrix $\Omega = (\omega_{uv})_{n \times n}$. Then, for any edge $(u, v) \in E$, its forest edge centrality $\text{FEC}(u, v)$ is*

$$\text{FEC}(u, v) = \frac{\omega_{uu} + \omega_{vv} - 2\omega_{uv}}{\omega_{uv}}. \qquad (3)$$

It has been proven that the upper bound of FEC for edge $(u, v)$ is $\text{FEC}(u, v) \leq d_u + d_v$, where $d_u$ and $d_v$ are the degrees of nodes $u$ and $v$, respectively. We find that both FNC and FEC can be expressed in terms of the entries of the forest matrix, which can be obtained by directly inverting the matrix $(I + L)$. This algorithm is referred to as EXACT (with its pseudocode provided in the appendix). While

straightforward, it has a time complexity of $O(n^3)$, which is imprac­tical for large-scale networks. Additionally, storing all entries of the forest matrix requires $O(n^2)$ space complexity. In practice, however, only the $n$ diagonal entries and the $m$ off-diagonal entries are nec­essary for our calculations. In the following sections, we introduce two novel approximation algorithms that achieve linear time and space complexity, making them scalable for large networks.

## 3 Algorithms for Estimating Entries of Forest Matrix on Undirected Graphs

Inspired by the forest theorem [8, 12], sampling-based methods have been widely employed to estimate the diagonal entries [40], trace [7] and column sum [39] of the forest matrix. In this section, we propose a novel variance reduction technique and introduce a new estimator for the entries of the forest matrix, leveraging the concept of isomorphic forests.

### 3.1 Existing Sample-Based Methods

Existing sample-based methods consist of two main steps: generat­ing a list of spanning forests uniformly, and estimating the entries based on each sampled forest.

*3.1.1 Wilson's algorithm and its extension.* Wilson's sample algo­rithm [44] and its extension [4] provides a method of sampling spanning trees and forests uniformly, based on loop-erased random walk [27]. It has been proven to sample a spanning forest in $O(n)$ time complexity [39], where $n$ is the number of nodes in the graph.

The procedure of the extension of Wilson's algorithm is detailed in [40]. Notably, in the implementation of the algorithm, we store each forest using an array next of length $n$, which records the next node for each node. Additionally, to quickly query the root of each node, we need an array root of length $n$. Therefore, each forest requires $O(n)$ space complexity. Although we eventually sample a list of forests, each forest is independent, and its contribution can be recorded after every sampling. Thus, the overall space complexity of the algorithm remains $O(n)$.

*3.1.2 Simple estimator.* The entry $\omega_{uv}$ represents the probability of node $u$ rooting at $v$ in a uniformly sampled spanning forest $\phi \in \mathcal{F}$. In the work in [40], according to the probability interpretation of $\omega_{uv}$, a simple unbiased estimator $\widehat{\omega}_{uv}(\phi)$ was used to estimate $\omega_{uv}$, defined as $\widehat{\omega}_{uv}(\phi) = \mathbb{I}_{\{r_\phi(u)=v\}}$, where $\mathbb{I}$ is a indicator function taking the value 1 when $u$ is rooted at $v$ and 0 otherwise. $\widehat{\omega}_{uv}(\phi)$ is an unbiased estimator of $\widehat{\omega}_{uv}$ with the variance of $\omega_{uv} - \omega_{uv}^2$. The algorithm that employs this estimator is referred to as SCF.

*3.1.3 Neighbor-based estimation method.* Although $\omega_{uv}$ is an un­biased estimator, its variance is quite large, leading to significant errors in SCF. Research [41] proposed the algorithm SFQPʟᴜs with a new estimator. For any node $u$ and $v$, it considers whether $v$ and its neighbors can potentially become the root of $u$. Specifi­cally, the new neighbor-based estimators for $\omega_{uv}$ and $\omega_{uu}$ are de­fined as $\widetilde{\omega}_{uv}(\phi) = \frac{1}{2+d_v}(\widehat{\omega}_{uv}(\phi) + \sum_{w \in N_v} \widehat{\omega}_{uw}(\phi))$, and $\widetilde{\omega}_{uu}(\phi) = \frac{1}{1+d_u}(1 + \sum_{w \in N_u} \widehat{\omega}_{uw}(\phi))$. It has been proven that the estimator $\widetilde{\omega}_{uv}(\phi)$ is an unbiased estimator of $\omega_{uv}$ and has a reduced variance compared to that of $\widehat{\omega}_{uv}(\phi)$.

### 3.2 Isomorphic Forest Group and New Estimators on Undirected Graph

Although existing works have established the relationship between the entries of the forest matrix and spanning forests and have pro­posed sampling algorithms to estimate them, there are still unique properties and information on undirected graphs that require fur­ther exploration. In this subsection, we introduce the concept of an isomorphic forest group. Then, based on this concept we propose new estimators for the entries of the forest matrix, which have lower variance than those in previous works.

We begin by introducing definitions related to the isomorphic forest group. Two trees $\tau_1$ and $\tau_2$ are considered isomorphic, de­noted $\tau_1 \sim \tau_2$, if they have the same set of nodes and edges, differing only in their root nodes. For a given tree, the number of its iso­morphic trees is equal to the number of its nodes, with each node serving as a distinct root. Similarly, two forests $\phi_1, \phi_2$ are defined as isomorphic if each tree within them has a one-to-one correspon­dence of isomorphism, denoted as $\phi_1 \sim \phi_2$. These forests together form an isomorphic forest group.

Figure 1 illustrates a spanning forest $\phi$ and all its isomorphic forests on a toy graph $\mathcal{G}$ with 7 nodes and 11 edges. The roots of trees are marked in yellow. For the forest $\phi$, we denote the tree containing nodes $\{4, 5, 6\}$ as $\tau_1$, and the tree containing $\{1, 2, 3, 7\}$ as $\tau_2$. There are three trees that are isomorphic to $\tau_1$, marked in blue, and four trees that are isomorphic to $\tau_2$. Therefore, the number of forests isomorphic to $\phi$ is $3 \times 4 = 12$, and all isomorphic forests are listed in Figure 1.

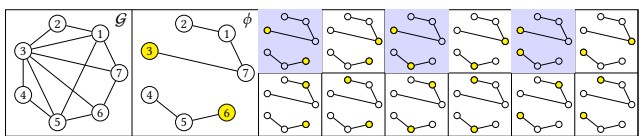

**Figure 1: A toy graph $\mathcal{G}$, a spanning forest $\phi$ and its isomor­phic forests.**

We now examine the process of estimating the forest matrix en­try $\omega_{uv}$. Consider a sampled forest $\phi$ consisting of $k$ trees, denoted as $\phi = \{\tau_1, \tau_2, \cdots, \tau_k\}$. Suppose nodes $u$ and $v$ are in the same tree $\tau_m$, but $v$ is not the root. In this situation, the simple estimator $\widehat{\omega}_{uv}(\phi)$ would fail to account for the contribution of such configu­rations. However, since the extension of Wilson's algorithm returns a uniform spanning forest from $\mathcal{F}$, the probability of obtaining a tree isomorphic to $\tau_m$, with $v$ as the root instead, is $\frac{1}{|T_\phi(u)|}$.

If we sample a forest in which $u$ and $v$ appear in the same tree, we can consider that this forest contributes $\frac{1}{|T_\phi(u)|}$ to $\omega_{uv}$ for each node $v$ in the connected component. Based on this, we propose a new estimator $\ddot{\omega}_{uv}(\phi)$ for $\omega_{uv}$, expressed as

$$\ddot{\omega}_{uv}(\phi) = \mathbb{E}\{\widehat{\omega}_{uv}(\phi_0)|\phi_0 \sim \phi\} = \frac{\mathbb{I}_{\{r_\phi(u)=r_\phi(v)\}}}{|T_\phi(u)|}.$$

For the case $u = v$, $\ddot{\omega}_{uu}(\phi) = \frac{1}{|T_\phi(u)|}$. In practice, using this esti­mator, sampling each forest $\phi$ is equivalent to obtaining $\prod_{\tau_i \in \phi} |V_{\tau_i}|$ distinct forests. Intuitively, with the number of samples remaining

constant, this estimator is expected to yield more effective results. Theorem 3.1 (whose proof is given in the appendix) demonstrates that $\ddot{\omega}_{uv}$ is an unbiased estimator with reduced variance.

**THEOREM 3.1.** *For a spanning forest $\phi \in \mathcal{F}$ and $u, v \in V$, $\ddot{\omega}_{uv}(\phi)$ is an unbiased estimator of $\omega_{uv}$, with a variance less than that of $\widehat{\omega}_{uv}(\phi)$.*

## 3.3 Combined Variance Reduced Estimator

We observe that the variance reduction technique for undirected graphs described in Section 3.2 and the neighbor-based sampling method from Section 3.1.3 reduce variance from different perspectives. For nodes $u, v \in V$, consider the estimation process of $\omega_{uv}$ using different estimators. For the simple estimator $\widehat{\omega}_{uv}$, only the cases where node $u$ is rooted at $v$ are considered. For the neighbor-based estimator $\widetilde{\omega}_{uv}$, forests where the root of $u$ is either $v$ or one of the neighbors of $v$ are considered. For the estimator $\ddot{\omega}_{uv}$, all forests where $u$ and $v$ belong to the same connected component are taken into account. These two estimators ($\widetilde{\omega}_{uv}$ and $\ddot{\omega}_{uv}$) intuitively increase the number of considered samples from different perspectives and have been proven to reduce variance independently. Therefore, we can combine them in a simple yet non-trivial way to construct a new estimator for approximating the entries of the forest matrix on undirected graphs.

For a spanning forest $\phi$ and two nodes $u, v$, we consider nodes such that node $w$ is in the same connected component as node $u$, and $w$ is either $v$ or a neighbor of $v$. This intuitively suggests that more sampled forests contribute to the calculation of $\omega_{uv}$. Specifically, our new estimators $\overline{\omega}_{uv}(\phi)$ and $\overline{\omega}_{uu}(\phi)$ are defined as follows:

$$\overline{\omega}_{uv}(\phi) = \frac{1}{2 + d_v}\left(\ddot{\omega}_{uv}(\phi) + \sum_{w \in N_v} \ddot{\omega}_{uw}(\phi)\right)$$

$$= \frac{1}{2 + d_v}\frac{1}{|T_\phi(u)|}\left(\mathbb{I}_{\{r_\phi(u) = r_\phi(v)\}} + \sum_{w \in N_v}\mathbb{I}_{\{r_\phi(u) = r_\phi(w)\}}\right),$$

$$\overline{\omega}_{uu}(\phi) = \frac{1}{1 + d_u}\left(1 + \sum_{w \in N_u}\ddot{\omega}_{uw}(\phi)\right)$$

$$= \frac{1}{1 + d_u}\left(1 + \frac{1}{|T_\phi(u)|}\sum_{w \in N_u}\mathbb{I}_{\{r_\phi(u) = r_\phi(w)\}}\right),$$

**THEOREM 3.2.** *For a spanning forest $\phi \in \mathcal{F}$ and $u, v \in V$, $\overline{\omega}_{uv}(\phi)$ is an unbiased estimator of $\omega_{uv}$, with variances less than that of $\widetilde{\omega}_{uv}(\phi)$. $\overline{\omega}_{uu}(\phi)$ is an unbiased estimator of $\omega_{uu}$, with variances less than that of $\widetilde{\omega}_{uu}(\phi)$.*

Theorem 3.2 demonstrates that $\overline{\omega}_{uv}$ and $\overline{\omega}_{uu}$ are unbiased estimators of $\omega_{uv}$ and $\omega_{uu}$, with variances less than the estimators in Section 3.1.3, respectively, as proved in the appendix. It is noteworthy that computing the number of nodes $k$ that meet the requirements for each edge $(u, v)$ cannot be accomplished in $O(1)$ time. For $\widetilde{\omega}_{uv}(\phi)$, counting requires only checking through the adjacency matrix whether the root $k$ of $u$ is a neighbor of $v$; for $\ddot{\omega}_{uv}(\phi)$, it suffices to determine whether $u$ and $v$ belong to the same connected component (i.e., whether their roots are identical). Both of the above operations take $O(1)$ time for each entry in every sampling iteration, resulting in an overall algorithm complexity of $O((n+m)l)$. In contrast, our combined algorithm necessitates evaluating the size

of the intersection between the connected component set of $u$ and the neighbor set of $v$. Such an operation takes $O(d_v)$ time.

## 3.4 Approximation Algorithm for FNC

Using the estimator for the diagonal entries, we can calculate the FNC of nodes. Specifically, assume that we sample $l$ forests to form a list $\mathcal{L}$, and define $\eta_u = \frac{1}{\omega_{uu}}$ and the estimator $\bar{\eta}_u(\mathcal{L}) = \frac{l}{\sum_{\phi \in \mathcal{L}} \overline{\omega}_{uu}(\phi)}$ for each node $u \in V$. Using the Chernoff bound [13], we provide a proper choice of the sample number $l$ and establish an $(\epsilon, \delta)$-approximation of $\bar{\eta}_u(\mathcal{L})$ in Theorem 3.3 (the proof is provided in the appendix).

**THEOREM 3.3.** *For any $\epsilon, \delta \in (0, 1)$, if $l$ is chosen obeying $l = \left\lceil\left(\frac{2(1+\epsilon)}{3\epsilon} + \frac{(1+\epsilon)^2}{4\epsilon^2}\right)\ln\frac{2}{\delta}\right\rceil$, the following inequalities hold with probability at least $1 - \delta$:*

$$\frac{1}{1+\epsilon}\omega_{uu} \leq \frac{1}{l}\sum_{\phi \in \mathcal{L}}\overline{\omega}_{uu}(\phi) \leq \frac{2+\epsilon}{1+\epsilon}\omega_{uu}.$$

*Then, the approximation $\bar{\eta}_u$ of $\eta_u$ satisfies the following relation:*

$$(1 - \epsilon)\eta_u \leq \bar{\eta}_u(\mathcal{L}) \leq (1 + \epsilon)\eta_u.$$

Compared to $\widetilde{\omega}_{uv}(\phi)$ and $\widetilde{\omega}_{uu}(\phi)$, our proposed estimators, $\overline{\omega}_{uv}(\phi)$ and $\overline{\omega}_{uu}(\phi)$, exhibit lower variance, thus requiring fewer expected samples to achieve the same error guarantee. To further accelerate the sampling process while maintaining the error bounds, we introduce the empirical Bernstein inequality [3], which allows us to tighten the theoretical bound without sacrificing accuracy.

**LEMMA 3.4.** *Let $X_1, X_2, \cdots, X_n$ be $n$ independent random variables satisfying $0 \leq X_i \leq M$. If we denote $\overline{X}$ and $X_{var}$ as the empirical mean and the empirical variance of $X_i$, then we have:*

$$\mathbb{P}\left\{|\overline{X} - \mathbb{E}(X)| \geq f(n, X_{var}, M, \delta)\right\} \leq \delta,$$

*where*

$$f(n, X_{var}, M, \delta) = \sqrt{\frac{2X_{var}\log(3/\delta)}{n}} + \frac{3M\log(3/\delta)}{n}.$$

Lemma 3.4 presents the empirical variance of random variables, which is initially unknown but can be effectively tracked throughout the sampling process. Specifically, we continue utilizing the Chernoff bound for the necessary number of sampled forests to guarantee that there is no loss of theoretical accuracy. By applying the Bernstein inequality, with each forest sampled, we update the empirical variance for each entry accordingly with each sampled forest. If the empirical errors of all entries are less than the desired error threshold, we terminate the sampling process. We introduce Algorithm IFGN to estimate the FNC for each node.

Algorithm 1 named IFGN, utilizes the isomorphic forest group (IFG) on undirected graphs along with neighbor (N) information to estimate FNC. According to our analysis, the time complexity of Algorithm 1 is $O(nl\bar{d})$, where $l$ is the number of predetermined number of samples, given by $l = \left\lceil\left(\frac{2(1+\epsilon)}{3\epsilon} + \frac{(1+\epsilon)^2}{4\epsilon^2}\right)\ln\frac{2}{\delta}\right\rceil$. The space complexity of the algorithm is $O(n)$. Although IFGN has a theoretically longer running time than using the variance reduction techniques discussed in Section 3.1.3 and Section 3.2 separately (both having a time complexity of $O(nl)$), we find that its high accuracy allows for earlier termination of sampling with relatively

**Algorithm 1:** IFGN($\mathcal{G}, \epsilon$)

**Input**  : $\mathcal{G}$: an undirected unweighted graph,
            $\epsilon$: an error parameter.
**Output**  : $\bar{\eta}$: a vector of the approximate values of FNC for each node
            $u \in V$.

1 **Initialize:** $l \leftarrow \left\lceil \left( \frac{2(1+\epsilon)}{3\epsilon} + \frac{(1+\epsilon)^2}{4\epsilon^2} \right) \log \frac{2}{\delta} \right\rceil$, $\bar{\eta}_u \leftarrow 0$ for $u \in V$. **for**
    $k = 1, 2, \cdots, l$ **do**
2 $\quad$ root, next $\leftarrow$ Wilson($\mathcal{G}$)
3 $\quad$ **foreach** $u \in V$ **do**
4 $\quad\quad$ $r \leftarrow$ root$_u$
5 $\quad\quad$ component$_r \leftarrow$ component$_r + 1$
6 $\quad$ **foreach** $u \in V$ **do**
7 $\quad\quad$ $t_u \leftarrow t_u + \frac{1}{(1+d_u)}$, $r \leftarrow$ root$_u$
8 $\quad\quad$ **for** $k \in N_u$ **do**
9 $\quad\quad\quad$ **if** $r =$ root$_k$ **then**
10 $\quad\quad\quad\quad$ $t_u \leftarrow t_u + \frac{1}{(1+d_u)} \frac{1}{\text{component}_r}$
11 $\quad\quad$ $\bar{\eta}_u \leftarrow \frac{k}{t_u}$
12 $\quad$ **if** $f(k, \text{var}(\bar{\eta}_u), \frac{2}{1+d_u}, \frac{1}{n}) \leq \epsilon$ *for all* $u \in V$ **then**
13 $\quad\quad$ break

14 **return** $\bar{\eta}$

fewer iterations due to the utilizing of Bernstein inequality, resulting in an overall running time that is faster than either method used individually, as shown in our experiments.

## 4 Novel Interpretation and Approximation Algorithm for FEC

We have proposed estimators for both the diagonal and off-diagonal entries of the forest matrix, along with a fast algorithm for estimating FNC. However, the structure of FEC is more complex, as it involves combinations of these entries.

In this section, we propose a novel physical interpretation of FEC. Based on this, we design a new estimator and a scalable algorithm FECE to approximate FEC.

### 4.1 Review for FEC Definition

The definition of FEC inherently possesses physical interpretation. As outlined in Definition 2.2, its value can be estimated through statistical methodologies. Specifically, for a list of uniformly sampled spanning forests, we can record the number of forests where $v$ is rooted at $u$. Additionally, we notice that during the execution of the extension of Wilson's algorithm, we maintained the array next, which conveniently allows us to determine whether the edge $(u, v)$ belongs to the forest. Define $\theta_{uv} = \text{FEC}(u, v)$, and assume that we sample $l$ forests to form a list $\mathcal{L}$. Then we can directly design an estimator $\widehat{\theta}_{uv}(\mathcal{L})$ as:

$$\widehat{\theta}_{uv}(\mathcal{L}) = \frac{\sum_{\phi \in \mathcal{L}} \mathbb{I}_{\{r_\phi(u)=v, \text{next}_v=u\}} |T_\phi(u)|}{\sum_{\phi \in \mathcal{L}} \mathbb{I}_{\{\text{root}_v=u\}}}$$

This formulation provides a straightforward method for estimating the FEC based on its definition. While simple, the method can yield significant errors in actual sampling because it only accounts for the result when there is an edge $(u, v)$ in the forest while other forests are discarded. When the number of samples is relatively low, the accuracy of this estimation is compromised.

### 4.2 New Physical Interpretation and Estimator

Since FEC can be expressed in terms of entries of the forest matrix, we proceed with further derivation from the expression in Equation (3):

$$\theta_{uv} = \frac{\omega_{uu} + \omega_{vv} - 2\omega_{uv}}{\omega_{uv}} = \frac{|\mathcal{F}_{uu}| - |\mathcal{F}_{vu}| + |\mathcal{F}_{vv}| - |\mathcal{F}_{uv}|}{|\mathcal{F}_{uv}|} \quad (4)$$

We observe that $\mathcal{F}_{uu}$ represents the set of all forests where $u$ is the root, and $\mathcal{F}_{vu}$ is the set of forests where $v$ is rooted at $u$. Clearly, $\mathcal{F}_{vu}$ is a proper subset of $\mathcal{F}_{uu}$, denoted as $\mathcal{F}_{vu} \subsetneq \mathcal{F}_{uu}$. Therefore, we have $|\mathcal{F}_{uu}| - |\mathcal{F}_{vu}| = \mathcal{F}_{uu} \setminus \mathcal{F}_{vu}$. Likewise, it follows that $|\mathcal{F}_{vv}| - |\mathcal{F}_{uv}| = |\mathcal{F}_{vv} \setminus \mathcal{F}_{uv}|$.

We define $\mathcal{S}_{u \setminus v} = \mathcal{F}_{uu} \setminus \mathcal{F}_{vu}$ and $\mathcal{S}_{v \setminus u} = \mathcal{F}_{vv} \setminus \mathcal{F}_{uv}$. Through our definition, we find that the set $\mathcal{S}_{u \setminus v}$ represents the collection of forests where $u$ is the root and $v$ is not in the same connected component as $u$. In this way, even if $u$ and $v$ are in different connected components, we can still consider that the forest contributes to the significance of edge $(u, v)$.

This conclusion might be difficult to comprehend: why is the size of the connected component containing the edge $(u, v)$ equal to the sum of the sizes of the sets $\mathcal{S}_{u \setminus v}$ and $\mathcal{S}_{v \setminus u}$? By combining the derivations and proofs from [5], we find that every forest $\phi$ that has $u$ as the root and includes the edge $(u, v)$ can be mapped to $|T_\phi(u)|$ isomorphic forests that do not contain the edge $(u, v)$. Moreover, these forests precisely constitute the union of the sets $\mathcal{S}_{u \setminus v}$ and $\mathcal{S}_{v \setminus u}$.

Figure 2 illustrates the specific process of this mapping. For a sampled forest $\phi$ containing the edge $(v, u)$, it contributes to the size of the connected component, which effectively means that each node within the component contributes a value of 1.

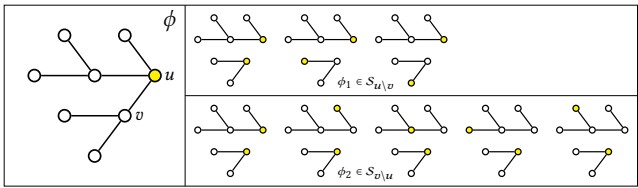

**Figure 2: A spanning forest $\phi$ and its mapped set.**

Using the new physical interpretation, for a spanning forest where $u$ and $v$ are in different trees, we record the number of times $u$ and $v$ being the root to estimate $|\mathcal{F}_{uu} \setminus \mathcal{F}_{vu}|$ and $|\mathcal{F}_{vv} \setminus \mathcal{F}_{uv}|$. Assume we sample $l$ forests to form a list $\mathcal{L}$. For each edge $(u, v) \in E$, define $\mathcal{H}_{uv}(\mathcal{L}) = \frac{|\mathcal{S}_{u \setminus v} + \mathcal{S}_{v \setminus u}|}{|\mathcal{F}|} |\mathcal{L}|$ and $\mathcal{K}_{uv}(\mathcal{L}) = \frac{|\mathcal{F}_{uv}|}{|\mathcal{F}|} |\mathcal{L}|$. Then, we define the estimator $\widetilde{\mathcal{H}}_{uv}(\mathcal{L})$ and $\widetilde{\mathcal{K}}_{uv}(\mathcal{L})$ as

$$\widetilde{\mathcal{H}}_{uv}(\mathcal{L}) = \sum_{\phi \in \mathcal{L}} (\mathbb{I}_{\{r_\phi(u) \neq r_\phi(v), r_\phi(u)=u\}} + \mathbb{I}_{\{r_\phi(u) \neq r_\phi(v), r_\phi(v)=v\}}),$$

$$\widetilde{\mathcal{K}}_{uv}(\mathcal{L}) = \sum_{\phi \in \mathcal{L}} \mathbb{I}_{\{r_\phi(v)=u\}}.$$

**Theorem 4.1.** *For edge* $(u, v) \in E$ *and the list* $\mathcal{L}$ *of* $l$ *sampled spanning forests,* $\widetilde{\mathcal{H}}_{uv}(\mathcal{L})$ *and* $\widetilde{\mathcal{K}}_{uv}(\mathcal{L})$ *are unbiased estimators of* $\mathcal{H}_{uv}(\mathcal{L})$ *and* $\mathcal{K}_{uv}(\mathcal{L})$*, respectively.*

We can still apply the variance reduction technique for undirected graphs in Section 3.2. Specifically, for the estimation of $\mathcal{K}_{uv}(\mathcal{L})$, whenever a sample places $u$ and $v$ within the same connected component, we increment the value by $\frac{1}{|T_\phi(u)|}$. Similarly, for $\mathcal{H}_{uv}(\mathcal{L})$, we avoid checking whether $u$ or $v$ is the root. Instead, we add $\frac{1}{|T_\phi(u)|} + \frac{1}{|T_\phi(v)|}$ for all forests where $u$ and $v$ are not in the same connected component. Then, we can construct the estimator $\overline{\mathcal{H}}_{uv}(\mathcal{L})$ and $\overline{\mathcal{K}}_{uv}(\mathcal{L})$ as

$$\overline{\mathcal{H}}_{uv}(\mathcal{L}) = \sum_{\phi \in \mathcal{L}} \mathbb{I}_{\{r_\phi(u) \neq r_\phi(v)\}} \left( \frac{1}{|T_\phi(u)|} + \frac{1}{|T_\phi(v)|} \right),$$

$$\overline{\mathcal{K}}_{uv}(\mathcal{L}) = \sum_{\phi \in \mathcal{L}} \mathbb{I}_{\{r_\phi(v) = r_\phi(u)\}} \frac{1}{|T_\phi(u)|}.$$

**Theorem 4.2.** *For edge* $(u, v) \in E$ *and the list* $\mathcal{L}$ *of* $l$ *sampled spanning forests,* $\overline{\mathcal{H}}_{uv}(\mathcal{L})$ *and* $\overline{\mathcal{K}}_{uv}(\mathcal{L})$ *are unbiased estimators of* $\mathcal{H}_{uv}(\mathcal{L})$ *and* $\mathcal{K}_{uv}(\mathcal{L})$*, respectively.*

### 4.3 Algorithm Design and Analysis

According to Equation (4) and Theorem 4.2, we have $\theta_{uv} = \frac{\mathbb{E}(\overline{\mathcal{H}}_{uv}(\mathcal{L}))}{\mathbb{E}(\overline{\mathcal{H}}_{uv}(\mathcal{L}))}$. Then, for each edge $(u, v) \in E$, we can build a new estimator for the FEC as $\bar{\theta}_{uv}(\mathcal{L}) = \frac{\overline{\mathcal{H}}_{uv}(\mathcal{L})}{\overline{\mathcal{K}}_{uv}(\mathcal{L})}$. Below, we demonstrate that the estimator $\bar{\theta}_{uv}$, with a proper choice of sample number $l$, can be used to approximate FEC. Specifically, we establish an $(\epsilon, \delta)$-approximation of this estimator using Hoeffding's inequality.

**Lemma 4.3 (Hoeffding's inequality [23]).** *Let* $x_1, x_2, \cdots, x_n$ *be* $l$ *independent random variables satisfying* $a \leq x_i \leq b$ *for all* $i = 1, 2, \cdots, n$*. Let* $x = \frac{1}{l} \sum_{i=1}^{l} x_i$*. Then for any* $\epsilon > 0$*,*

$$\mathbb{P}\{|x - \mathbb{E}(x)| \geq \epsilon\} \leq 2 \exp\left(-\frac{2l\epsilon^2}{(b-a)^2}\right).$$

**Theorem 4.4.** *For any edge* $(u, v) \in E$ *with* $\omega_{uv} \geq \sigma$*, and parameters* $\epsilon, \sigma, \delta \in (0, 1)$*, if* $l$ *is chosen obeying* $l = \left\lceil \frac{(2+\epsilon)^2}{2\epsilon\sigma^2} \ln \frac{2}{\delta} \right\rceil$*, then the following inequalities holds with probability at least* $1 - \delta$*:*

$$\mathbb{P}\{|\overline{\mathcal{H}}_{uv}(\mathcal{L}) - \mathcal{H}_{uv}(\mathcal{L})| \leq |\mathcal{L}| \frac{\sigma\epsilon(d_u + d_v)}{2 + \epsilon}\} < \delta, \quad (5)$$

$$\mathbb{P}\{|\overline{\mathcal{K}}_{uv}(\mathcal{L}) - \mathcal{K}_{uv}(\mathcal{L})| \leq |\mathcal{L}| \frac{\sigma\epsilon}{2 + \epsilon}\} < \delta. \quad (6)$$

*Then, the approximation* $\bar{\theta}_{uv}(\mathcal{L})$ *of* FEC *satisfies the following relation:*

$$\theta_{uv} - (d_u + d_v)\epsilon \leq \bar{\theta}_{uv}(\mathcal{L}) \leq \theta_{uv} + (d_u + d_v)\epsilon. \quad (7)$$

Using the estimator $\bar{\theta}_{uv}$ for each edge, we propose the algorithm FECE (Forest Edge Centrality Estimation) to estimate the FEC of edges directly.

Given a graph $\mathcal{G}$ and an error parameter $\epsilon$, FECE first computes the expected sample number $l = \left\lceil \frac{(2+\epsilon)^2}{2\epsilon\sigma^2} \ln \frac{2}{\delta} \right\rceil$ (Line 1). Then, in each sampling iteration, FECE invokes the extension of Wilson's algorithm to obtain a spanning forest and computes the size of the

---

**Algorithm 2:** FECE$(\mathcal{G}, \epsilon)$

**Input** : $\mathcal{G}$: an undirected unweighted graph,
$\epsilon$: an error parameter.

**Output** : $\bar{\theta}$: a vector of the approximate values of FEC for each edge $(u, v) \in E$.

1 **Initialize**: $l \leftarrow \left\lceil \frac{(2+\epsilon)^2}{2\epsilon\sigma^2} \ln \frac{2}{\delta} \right\rceil$.

2 **for** $k = 1, 2, \cdots, l$ **do**
3    root, next $\leftarrow$ Wilson$(\mathcal{G})$
4    **foreach** $u \in V$ **do**
5      $r \leftarrow$ root$_u$,
6      component$_r \leftarrow$ component$_r + 1$
7    **foreach** $(u, v) \in E$ **do**
8      $r \leftarrow$ root$_u$, $w \leftarrow$ root$_v$
9      **if** $r = w$ **then**
10        $\overline{\mathcal{K}}_{uv} \leftarrow \overline{\mathcal{K}}_{uv} + \frac{1}{\text{component}_r}$
11      **else if** $r \neq w$ **then**
12        $\overline{\mathcal{H}}_{uv} \leftarrow \overline{\mathcal{H}}_{uv} + \frac{1}{\text{component}_r} + \frac{1}{\text{component}_w}$
13    $\bar{\theta}_{uv} \leftarrow \overline{\mathcal{H}}_{uv} / \overline{\mathcal{K}}_{uv}$ for $(u, v) \in E$
14    **if** $f(k, \text{var}(\bar{\theta}_{uv}), \frac{1}{4}, \frac{1}{n}) \leq \epsilon$ for all $(u, v) \in E$ **then**
15      break

16 **return** $\bar{\theta}$

---

connected component that includes each node. (Lines 3-6). For each edge $(u, v)$, FECE checks whether $u$ and $v$ are in the same connected component, updates $\overline{\mathcal{K}}_{uv}$ and $\overline{\mathcal{H}}_{uv}$, and finally calculates the new $\bar{\theta}_{uv}$ (Lines 7-13). Bernstein inequality is used to terminate sampling early when the empirical variance is below a specified threshold (Lines 14-15). The complexity of Algorithm 2 is $O\left(\frac{n(2+\epsilon)^2}{2\epsilon\sigma^2} \ln \frac{2}{\delta}\right)$.

## 5 Experiments

In this section, we present experimental results for real-world networks to demonstrate the accuracy and efficiency of our approximation algorithms for FNC and FEC.

### 5.1 Experimental Settings

**Datasets.** We conduct our experiments on real-world networks from Koblenz Network Collection [26], SNAP [28], and Network Data Repository [35]. This includes a wide range of datasets, including collaboration networks (GrQc), social networks (YouTube), and citation networks (US Patents). We select datasets with varying edge-to-node ratios, as the running time of certain algorithms is influenced by this factor. Statistics for the networks are given in Table 1, sorted in ascending order by the number of nodes.

**Environment.** All experiments were conducted on a Linux server with 36-core 2.10GHz Intel(R) Xeon(R) Platinum 8352V CPU and 256GB of RAM. We implemented all the algorithms in Julia.

**Algorithms.** We use the algorithm Exact (Section 2.4) to assess the accuracy and efficiency of other algorithms on small-scale networks. For the computation of FNC, we implement two algorithms as our baseline: SCF (Section 3.1.2) uses the simple estimator, while SFQPlus (Section 3.1.3) employs the neighbor information to reduce variance. Our proposed algorithm IFGN (Section 4.3) combines the variance reduction technique on undirected graphs and

**Table 1: Datasets used in experiments**

| Type | Dataset | $n$ | $m$ | $m/n$ |
|---|---|---|---|---|
| | Bio-CE-LC | 1,387 | 1,648 | 1.18 |
| | Hamsterster | 2,000 | 16,097 | 8.05 |
| Small and Medium Network | Facebook | 4,039 | 88,234 | 21.85 |
| | GrQc | 4,158 | 13,422 | 3.23 |
| | WebBase | 16,062 | 25,593 | 1.59 |
| | Gnutella | 22,687 | 54,705 | 2.41 |
| | Youtube | 1,134,890 | 2,987,624 | 2.63 |
| Large Network | US Patents | 3,774,768 | 16,518,948 | 4.38 |
| | Dblp | 5,624,219 | 12,282,059 | 2.18 |
| | CentralUSA | 14,081,816 | 33,866,826 | 2.41 |

the approach based on neighbor information in a simple yet non-trivial way. Meanwhile, as our algorithm IFGN incorporates two optimization techniques, we also implement an ablation method called IFG, which solely utilizes the variance reduction technique on undirected graphs (with its pseudocode provided in the appendix). For FEC, we implement FECE and compare it with SCF and SFQPⱫᴜꜱ. The empirical Bernstein inequality in Lemma 3.4 is used to terminate sampling for all algorithms. The code implementation of all algorithms can be accessed directly through https://anonymous.4open.science/r/Forest-Based-Centrality-BD41.

**Parameters.** We set $\delta = 0.01$ for all algorithms. For FNC, we set the error parameter $\epsilon = 0.05$, and the lower bound for $l$ of IFGN is provided by Theorem 3.3. The analysis process for the sampling numbers of the two comparison methods is similar to that in Theorem 3.3. For FEC, the lower bound for $l$ of FECE is given in Theorem 4.4. Due to the difficulty in providing an error analysis for the comparison methods when calculating FEC, we use the same sampling number for all algorithms.

## 5.2 Accuracy Evaluation

We first evaluate the accuracy of our proposed algorithms. According to Equations (1) and (2.3), we use Exᴀᴄᴛ to compute the exact value of FNC of all nodes and FEC of all edges for small real-world networks. As the measure of accuracy, we use the mean relative error $\rho = \frac{1}{n} \sum_{i=1}^{n} |C_i - \widehat{C}_i|/C_i$, where $C$ is the exact value and $\widehat{C}$ is the approximate value. Six small datasets we selected include: Residence, Hamsterster, Adolescent, GrQc, WebBase and Gnutella, each containing fewer than $30,000$ nodes to ensure the feasibility of Exᴀᴄᴛ.

Our initial evaluation concentrates on the performance of our algorithms in estimating the FNC. There are four considered algorithms: SCF, SFQPⱫᴜꜱ, IFG, and IFGN, each with a specified error parameter $\epsilon = 0.05$. Results for these settings are reported in Figure 3.

Figure 3 indicates that all approximation algorithms for FNC are effective. IFG, which utilizes the variance reduction technique on undirected graphs, significantly reduces the estimation error compared to SCF, and achieves accuracy levels comparable to SFQPⱫᴜꜱ. We also observe that the combined algorithm IFGN demonstrates enhanced performance in estimating the FNC with a mean relative error of less than 0.005.

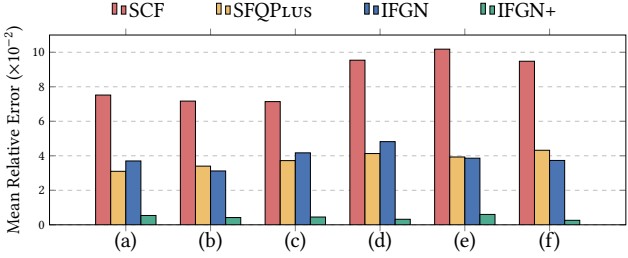

**Figure 3: Mean relative error of four algorithms for FNC on four real-world networks: Bio-CE-LC (a), Hamsterster (b), Facebook (c), GrQc (d), WebBase (e) and Gnutella (f).**

We then conduct experiments to evaluate the performance of algorithm FECE for estimating FEC. Since no theoretical error bounds are provided for the baseline algorithms, we use the number of samples $l$ as the parameter. For each algorithm, $l$ is set to 500, 1000, 1500, 2000, and 2500. In addition to the two baselines, we also modify the IFGN algorithm from Section 3.3 to create IFGN-E for comparison, which approximates the FEC by separately estimating the diagonal and off-diagonal entries. The scatter plots of mean relative error and running time for each approximation algorithm are reported in Figure 4.

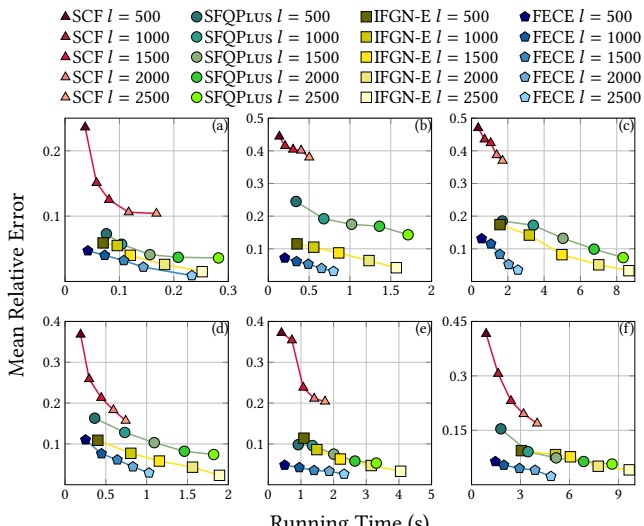

**Figure 4: Scatter plots of mean relative error and running time for each approximation algorithm for FEC on real-world networks: Bio-CE-LC (a), Hamsterster (b), Facebook (c), GrQc (d), WebBase (e) and Gnutella (f).**

Figure 4 illustrates that as the sample size $l$ increases, the error for each algorithm continuously decreases. SCF exhibits relatively large errors due to its high variance, which leads to inaccurate estimates for the off-diagonal entries. It is worth noting that the running time of IFGN-E was not as long as theoretically expected, owing to the use of the Bernstein inequality to terminate the sampling process early in the algorithm. For methods that directly

estimate the entries of the forest matrix (SCF, SFQPLUS and IFGN-E), additional errors may arise from recomputing FEC using the diagonal and non-diagonal entries. In contrast, FECE exhibits even better performance. When $l$ reaches 2000, FECE achieves a relative error of $\epsilon < 0.04$ for all six graphs. Under the same running time, FECE achieves the lowest relative error. Moreover, for the same error level, FECE has the shortest runtime. Such results further demonstrate the effectiveness of FECE in estimating FEC.

## 5.3 Efficiency Evaluation

The efficiency of the algorithms is also a crucial aspect of the evaluation, as many precise algorithms can be time-consuming when handling large networks. Algorithm EXACT requires over one hour to compute networks with more than 30,000 nodes. The method proposed in [24] based on the Johnson-Lindenstrauss lemma and fast SDDM Solver can only handle graphs with fewer than approximately one million nodes to ensure a relative error, which has been shown in [41] to be less efficient than sampling-based approaches. Our experiments are conducted on four large real-world networks, including Youtube (YT), US Patents (US), Dblp (DB) and CentralUSA (CU), each with over one million nodes and two million edges. The error parameter is set to be $\epsilon = 0.05$. The running time for estimating FNC and FEC is reported in Table 2.

**Table 2: The running time (seconds) of approximating FNC and FEC using SCF, SFQPLUS, IFG, IFGN and FECE.**

| Network | FNC | | | | FEC | | |
|---------|------|---------|-------|-------|------|---------|--------|
| | SCF | SFQPLUS | IFG | IFGN | SCF | SFQPLUS | FECE |
| YT | 159.7 | 161.5 | 142.2 | 104.3 | 205.4 | 191.3 | 178.2 |
| US | 989.9 | 1046 | 835.5 | 770.5 | 1695 | 1574 | 1385.6 |
| DB | 1066 | 994.9 | 901.5 | 951.4 | 1342 | 1258 | 1041 |
| CU | 1982 | 2179 | 1526 | 1359 | 2476 | 2381 | 2205 |

As shown in Table 2, all of the sample-based algorithms complete their approximations within one hour, even for large networks. The computation time for FEC is generally longer than that for FNC, especially in graphs with a higher average degree. Compared to SCF, our proposed algorithms IFGN and FECE complete the estimation of FNC and FEC faster while providing the same error guarantee. This improvement is due to Lemma 3.4, where the Bernstein inequality allows early termination of sampling when the empirical variance falls below a certain threshold. Even though the complexity of IFGN is $\bar{d}$ times that of SCF, its superior accuracy allows it to outperform other baselines in terms of overall running time.

## 6 Related Works

**Node centrality metrics.** Numerous concepts of node centrality have been developed. Some definitions, such as degree centrality [19], focus exclusively on local structural information. In contrast, measures like betweenness [18] and closeness [19] capture node significance within the context of the global structure. Nevertheless, these conventional methods depend entirely on shortest paths, limiting their capacity to represent intricate structural details [16, 24]. Furthermore, many centrality measures fail to effectively assess disconnected graphs. The forest closeness centrality

(FCC) introduced by Jin *et al.* [24] utilizes the concept of forest distance [10, 12, 31] to define a novel centrality measure, expressed as $C_{\mathsf{FCC}}(u) = \frac{n}{\sum_{v \in V \setminus \{u\}} \rho(u,v)}$, where $\rho(u,v)$ denotes the forest distance between $u$ and $v$. Due to its strong discriminative ability and applicability to disconnected graphs [6], FCC has garnered significant attention, and the algorithm for fast computation has been extensively studied [24, 40, 43].

**Edge centrality metrics.** In addition to node centrality, edge centrality metrics and related algorithms have garnered widespread research interest and attention in recent years. Existing edge centrality metrics include edge betweenness (EB) and spanning centrality (SC). EB was proposed by Freeman [18], defined as the probability that the shortest path between any pair of nodes passes through the edge. For an undirected graph $G = (V, E)$ the EB of edge $(u, v) \in E$ is defined as $C_{\mathsf{EB}}(u,v) = \sum_{s,t \in V} \frac{\sigma_{st}(u,v)}{\sigma_{st}}$, where $\sigma_{st}$ denotes the number of shortest paths between nodes $s$ and $t$, while $\sigma_{st}(u,v)$ denotes the number of shortest path between the node pair $s$ and $t$ including edge $(u, v)$. Teixeira and Monteiro proposed spanning edge betweenness [42], which is the probability that an edge is included in a uniformly chosen spanning tree of the graph. The SC of edge $(u, v)$ is defined as $C_{\mathsf{SC}}(u,v) = \frac{|\tau_{(u,v)}|}{|\tau|}$, where $\tau$ is the set of all spanning trees of $G$ and $\tau_{(u,v)}$ is the set of those spanning trees containing edge $(u, v)$. In more in-depth research, efficient approximation algorithms for SC on large-scale graphs have been proposed [22, 30, 45], gaining sustained attention for their scalability and practical relevance in handling complex network structures. Different centrality metrics take various factors into account, and each can differentiate and rank edges to some extent.

**Existing methods of computing forest matrix.** In recent years, considerable research has focused on developing fast computation methods for forest matrix. In the work in [24], the author utilized the fast SDDM solver [14, 38] for the calculation of the diagonal entries of the forest matrix. Research in [47] transformed the problem of minimizing polarization and disagreement in the Friedkin-Johnsen [20] model into the computation of quantities related to the forest matrix, achieving fast computation by leveraging the Johnson-Lindenstrauss [25] lemma and fast SDDM solver. Inspired by the forest theorem [8, 12], sample-based methods were widely researched to estimate the diagonal entries [40], trace [7] and column sum [39] of the forest matrix. Variance reduction techniques [34] are employed to optimize sampling algorithms, thereby enhancing their efficiency.

## 7 Conclusion

In this paper, we studied the problem of effectively approximating the forest node centrality (FNC) and forest edge centrality (FEC) on large graphs. Armed with novel variance reduction techniques and new physical interpretations, we proposed two scalable algorithms IFGN and FECE from different perspectives and provided theoretical error guarantees. Through comprehensive experiments on real-world datasets, we demonstrated that our algorithms, IFGN and FECE, significantly outperform existing methods in terms of both speed and accuracy.

In future works, we will explore further extensions of these centrality measures, as well as their applications in various network analysis tasks.

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

# A Appendix

In this section, we present eliminated pseudocode and proofs for theorems in the main text.

## A.1 Pseudocode of Algorithm EXACT

---

**Algorithm 3:** EXACT($\mathcal{G}$)

---

**Input** : $\mathcal{G}$: an undirected unweighted graph.
**Output** : FNC: the forest node centrality for each node
            FEC: the forest edge centrality for each edge
1 **Initialize:** $D \leftarrow$ Degree matrix of $\mathcal{G}$, $A \leftarrow$ Adjacency matrix of $\mathcal{G}$
2 Compute $\Omega = (I + D - A)^{-1}$
3 **foreach** $u \in V$ **do**
4     $\text{FNC}_u \leftarrow 1/\omega_{ii}$
5 **foreach** $(u, v) \in E$ **do**
6     $\text{FEC}_{(u,v)} \leftarrow \frac{\omega_{uu}+\omega_{vv}}{\omega_{uv}} - 2$
7 **return** FNC,FEC

---

## A.2 Proof of Theorem 3.1

**Proof.** For a spanning forest $\phi \in \mathcal{F}$ with $\phi = \{\tau_1, \tau_2, \cdots, \tau_k\}$ and two nodes $u, v \in V$, we consider two cases: (a) If $u$ and $v$ belong to different trees, we have $r_\phi(u) \neq r_\phi(v)$ and $\ddot{\omega}_{uv} = 0$. (b) If $u$ and $v$ belong to the same tree, without loss of generality, we suppose that both $u$ and $v$ are in $\tau_1$. Then, the probability of $r_\phi(u) = p$ is equal to that of $r_\phi(u) = q$ for different nodes $p$ and $q$ in $\tau_1$. Thus, $\ddot{\omega}_{uv}(\phi) = \frac{1}{|T_\phi(u)|}$.

The expectation of $\ddot{\omega}_{uv}(\phi)$ can be derived as

$$\mathbb{E}(\ddot{\omega}_{uv}(\phi)) = \mathbb{E}\left\{\mathbb{P}\{r_{\phi_0}(u) = v | \phi_0 \sim \phi\}\right\}$$
$$= \sum_{\phi_0} \mathbb{P}\{r_{\phi_0}(u) = v | \phi_0 \sim \phi\}\mathbb{P}\{\phi_0 \sim \phi\}$$
$$= \mathbb{P}\{r_{\phi_0}(u) = v\} = \omega_{uv},$$

which implies that $\ddot{\omega}_{uv}(\phi)$ is an unbiased estimator of $\omega_{uv}$.

Moreover, according to the total variance formula and the properties of conditional probabilities, we obtain

$$\text{Var}(\widehat{\omega}_{uv}(\phi)) = \text{Var}(\ddot{\omega}_{uv}(\phi)) + \mathbb{E}\{\text{Var}(\widehat{\omega}_{uv}(\phi_0)|\phi_0 \sim \phi)\}$$
$$\geq \text{Var}(\ddot{\omega}_{uv}(\phi)),$$

which completes the proof. $\square$

## A.3 Proof of Theorem 3.2

**Proof.** For the case $u \neq v$, using the relationship $\Omega(I + L) = I$, it follows that $e_u^\top \Omega(I + L)e_v = 0$, leading to $(1 + d_v)\omega_{uv} - \sum_{w \in N_v} \omega_{uw} = 0$. According to Theorem 3.1 we have

$$\mathbb{E}(\overline{\omega}_{uv}(\phi)) = \frac{1}{2 + d_v}\left(\mathbb{E}(\ddot{\omega}_{uv}(\phi)) + \sum_{w \in N_v} \mathbb{E}(\ddot{\omega}_{uw}(\phi))\right)$$
$$= \frac{1}{2 + d_v}(\omega_{uv} + (1 + d_v)\omega_{uv}) = \omega_{uv},$$

which implies that $\overline{\omega}_{uv}(\phi)$ is an unbiased estimator of $\omega_{uv}$. According to the properties of conditional probabilities, we have

$$\text{Var}(\widetilde{\omega}_{uv}(\phi)) = \text{Var}(\overline{\omega}_{uv}(\phi)) + \mathbb{E}\{\text{Var}(\widetilde{\omega}_{uv}(\phi_0)|\phi_0 \sim \phi)\}$$
$$\geq \text{Var}(\overline{\omega}_{uv}(\phi)).$$

For the case $u = v$, using $e_u^\top \Omega(I+L)e_u = 1$, we have $(1+d_u)\omega_{uu} - \sum_{w \in N_u} \omega_{uw} = 1$. Then, we obtain

$$\mathbb{E}(\overline{\omega}_{uu}(\phi)) = \frac{1}{1 + d_u}(1 + \sum_{w \in N_u} \mathbb{E}(\ddot{\omega}_{uw}(\phi)))$$
$$= \frac{1}{1 + d_u}(1 + (1 + d_u)\omega_{uv} - 1) = \omega_{uv},$$

which shows that $\overline{\omega}_{uu}(\phi)$ is an unbiased estimator of $\omega_{uu}$. Using the similar approach for the case $u \neq v$, we obtain $\text{Var}(\overline{\omega}_{uu}(\phi)) \leq \text{Var}((\widetilde{\omega}_{uu}(\phi)))$, which completes the proof. $\square$

## A.4 Chernoff Bound

**THEOREM A.1 (CHERNOFF BOUND [13]).** *Let $x_1, x_2, \cdots, x_l$ be $l$ independent random variables satisfying $|x_i - \mathbb{E}(x_i)| \leq M$ for all $i = 1, 2, \cdots, l$. Let $x = \frac{1}{l}\sum_{i=1}^{l} x_i$. Then, for any $\epsilon > 0$, we have*

$$\mathbb{P}\{|x - \mathbb{E}(x)| \geq \epsilon\} \leq 1 - 2\exp\left(-\frac{l\epsilon^2}{2(\text{Var}(x)l + M\epsilon/3)}\right).$$

## A.5 Proof of Theorem 3.3

**Proof.** Since the variance of $\overline{\omega}_{uu}$ is less than that of $\widetilde{\omega}_{uu}$ for each $u \in V$, using the theorem in [41] and the Chernoff bound, if $l$ is chosen obeying $l = \left\lceil\left(\frac{2(1+\epsilon)}{3\epsilon} + \frac{(1+\epsilon)^2}{4\epsilon^2}\right)\log\frac{2}{\delta}\right\rceil$, the relative error in [41] holds for $\overline{\omega}_{uu}$ at least $1 - \delta$, expressed as

$$\frac{1}{1+\epsilon}\omega_{uu} \leq \frac{1}{l}\sum_{\phi \in \mathcal{L}} \overline{\omega}_{uu}(\phi) \leq \frac{2+\epsilon}{1+\epsilon}\omega_{uu}.$$

Then the error of $\bar{\eta}_u$ can be bounded as follows:

$$\left|\frac{\bar{\eta}_u - \eta_u}{\eta_u}\right| \leq \left|\frac{(1+\epsilon)\eta_u - \eta_u}{\eta_u}\right| = \epsilon,$$

which completes the proof. $\square$

## A.6 Proof of Theorem 4.1

**Proof.** For $\widetilde{\mathcal{H}}_{uv}(\mathcal{L})$, we have $\mathbb{I}_{\{r_\phi(u)\neq r_\phi(v), r_\phi(u)=u\}} = \mathbb{I}_{\{r_\phi(u)=u\}} - \mathbb{I}_{\{r_\phi(v)=u\}}$ and $\mathbb{I}_{\{r_\phi(u)\neq r_\phi(v), r_\phi(v)=v\}} = \mathbb{I}_{\{r_\phi(v)=v\}} - \mathbb{I}_{\{r_\phi(u)=v\}}$. Since each forest $\phi \in \mathcal{L}$ is sampled uniformly,

$$\mathbb{E}(\widetilde{\mathcal{H}}_{uv}(\mathcal{L})) = |\mathcal{L}|((\mathbb{P}(r_\phi(u) = u) - \mathbb{P}(r_\phi(v) = u)$$
$$+ \mathbb{P}(r_\phi(v) = v) - \mathbb{P}(r_\phi(u) = v)))$$
$$= |\mathcal{L}|(\omega_{uu} + \omega_{vu} + \omega_{vv} - \omega_{uv}) = \mathcal{H}_{uv}(\mathcal{L}).$$

For $\widetilde{\mathcal{K}}_{uv}(\mathcal{L})$, we have $\mathbb{E}(\widetilde{\mathcal{K}}_{uv}(\mathcal{L})) = |\mathcal{L}|\mathbb{P}(r_\phi(v) = u) = \frac{|\mathcal{F}_{uv}|}{|\mathcal{F}|}|\mathcal{L}| = \mathcal{K}_{uv}(\mathcal{L})$, which completes the proof. $\square$

## A.7 Proof of Theorem 4.2

**Proof.** For $\overline{\mathcal{H}}_{uv}(\mathcal{L})$, we have $\mathbb{I}_{\{r_\phi(u)\neq r_\phi(v)\}} = 1 - \mathbb{I}_{\{r_\phi(u)=r_\phi(v)\}} = \mathbb{I}_{\{r_\phi(u)=r_\phi(u)\}} - \mathbb{I}_{\{r_\phi(u)\neq r_\phi(v)\}}$. Then,

$$\mathbb{E}(\overline{\mathcal{H}}_{uv}(\mathcal{L})) = |\mathcal{L}|(\mathbb{E}(\frac{\mathbb{I}_{\{r_\phi(u)\neq r_\phi(v)\}}}{|T_\phi(u)|}) + \mathbb{E}(\frac{\mathbb{I}_{\{r_\phi(u)\neq r_\phi(v)\}}}{|T_\phi(v)|}))$$
$$= |\mathcal{L}|(\omega_{uu} + \omega_{vu} + \omega_{vv} - \omega_{uv}) = \mathcal{H}_{uv}(\mathcal{L}).$$

For $\overline{\mathcal{K}}_{uv}(\mathcal{L})$, we have $\mathbb{E}(\overline{\mathcal{K}}_{uv}(\mathcal{L})) = |\mathcal{L}|\mathbb{E}(\frac{\mathbb{I}_{\{r_\phi(u)=r_\phi(v)\}}}{|T_\phi(u)|}) = |\mathcal{L}|\omega_{uv} = \mathcal{K}_{uv}(\mathcal{L})$, which completes the proof. $\square$

## A.8 Proof of Theorem 4.4

**Proof.** For any edge $(u, v) \in E$ and a spanning forest $\phi$, we have $\mathbb{I}_{\{r_\phi(u) \neq r_\phi(v)\}} \left( \frac{1}{|T_\phi(u)|} + \frac{1}{|T_\phi(v)|} \right) \leq 2$, and $\mathbb{I}_{\{r_\phi(u) = r_\phi(v)\}} \left( \frac{1}{|T_\phi(u)|} \right) \leq \frac{1}{2}$. Using the Hoeffding's inequality and choosing $l$ as previously specified, we can prove that the inequalities (5) and (6) holds with probability at least $1 - \delta$.

Then, the error of $\bar{\theta}_{uv}(\mathcal{L})$ can be bounded as follows:

$$|\bar{\theta}_{uv}(\mathcal{L}) - \theta_{uv}(\mathcal{L})| = \left| \frac{\overline{\mathcal{H}}_{uv}(\mathcal{L})}{\overline{\mathcal{K}}_{uv}(\mathcal{L})} - \frac{\mathcal{H}_{uv}(\mathcal{L})}{\mathcal{K}_{uv}(\mathcal{L})} \right|$$

$$= \frac{|\mathcal{H}_{uv}(\mathcal{L})(\overline{\mathcal{K}}_{uv}(\mathcal{L}) - \mathcal{K}_{uv}(\mathcal{L})) + \mathcal{K}_{uv}(\mathcal{H}_{uv}(\mathcal{L}) - \overline{\mathcal{H}}_{uv}(\mathcal{L}))|}{\overline{\mathcal{K}}_{uv}(\mathcal{L})\mathcal{K}_{uv}(\mathcal{L})}$$

$$\leq \frac{|\mathcal{H}_{uv}(\mathcal{L})(\overline{\mathcal{K}}_{uv}(\mathcal{L}) - \mathcal{K}_{uv}(\mathcal{L}))| + |\mathcal{K}_{uv}(\mathcal{H}_{uv}(\mathcal{L}) - \overline{\mathcal{H}}_{uv}(\mathcal{L}))|}{\overline{\mathcal{K}}_{uv}(\mathcal{L})\mathcal{K}_{uv}(\mathcal{L})}$$

$$= \frac{\frac{\mathcal{H}_{uv}(\mathcal{L})}{\mathcal{K}_{uv}(\mathcal{L})} \frac{\sigma\epsilon}{2+\epsilon} + \frac{\sigma\epsilon(d_u+d_v)}{2+\epsilon}}{\frac{\mathcal{K}_{uv}(\mathcal{L})}{|\mathcal{L}|} - \frac{\sigma\epsilon}{2+\epsilon}} \leq \frac{2(d_u + d_v)\frac{\sigma\epsilon}{2+\epsilon}}{\delta - \frac{\sigma\epsilon}{2+\epsilon}} = (d_u + d_v)\epsilon,$$

where the last inequality holds since $\frac{\mathcal{H}_{uv}(\mathcal{L})}{\mathcal{K}_{uv}(\mathcal{L})} = \theta_{uv} \leq d_u + d_v$ and $\frac{\mathcal{K}_{uv}(\mathcal{L})}{|\mathcal{L}|} = \omega_{uv} \geq \sigma$. This finishes the proof. □

## A.9 Pseudocode of Algorithm IFG

---

**Algorithm 4:** IFG($\mathcal{G}, \epsilon$)

---

**Input** : $\mathcal{G}$: an undirected unweighted graph,
$\epsilon$: an error parameter

**Output** : $\ddot{\eta}$: a vector of the approximate value of FNC for each node $u \in V$.

1 **Initialize**: $l \leftarrow \left\lceil \left( \frac{2(1+\epsilon)}{3\epsilon} + \frac{(1+\epsilon)^2}{4\epsilon^2} \right) \log \frac{2}{\delta} \right\rceil$, $\ddot{\eta}_u \leftarrow 0$ for $u \in V$.

2 **for** $k = 1, 2, \cdots, l$ **do**

3 $\quad$ root, next $\leftarrow$ Wilson($\mathcal{G}$)

4 $\quad$ **foreach** $u \in V$ **do**

5 $\quad\quad$ $r \leftarrow \text{root}_u$

6 $\quad\quad$ component$_r \leftarrow$ component$_r + 1$

7 $\quad$ **foreach** $u \in V$ **do**

8 $\quad\quad$ $t_u \leftarrow t_u + \frac{1}{\text{component}_u}$

9 $\quad\quad$ $\ddot{\eta}_u \leftarrow \frac{k}{t_u}$

10 $\quad$ **if** $f(k, \text{var}(\ddot{\eta}_u), 1, \frac{1}{n}) \leq \epsilon$ for all $u \in V$ **then**

11 $\quad\quad$ break

12 **return** $\ddot{\eta}$

---

