# OpenReview forum: "Scalable Algorithms for Forest-Based Centrality on Large Graphs"
_ACM.org/TheWebConf/2025/Conference — WWW 2025 Oral_

### Official Review · Reviewer_4D6b · 2024-11-13

**Novelty:** 3
**Technical Quality:** 4

**Review:**

Summary

The paper presents scalable algorithms, IFGN and FECE, for approximating forest-based centrality measures—Forest Node Centrality (FNC) and Forest Edge Centrality (FEC)—on large undirected graphs. These methods employ variance reduction techniques and are evaluated across various real-world networks, showing improvements in both efficiency and accuracy.


Strengths

S1. The paper addresses computational challenges in applying forest-based centrality to large, disconnected graphs, a relevant problem in graph-based data analysis.

S2. It provides rigorous theoretical foundations, including formal proofs and error guarantees for the proposed estimators.

S3. A diverse set of experiments demonstrates the methods’ efficiency and scalability on large networks.


Weaknesses

W1. The paper presents incremental advancements with limited novelty relative to prior work in variance reduction and forest centrality.

W2. Presentation is unclear, particularly in the introduction, where the motivation for the study is weak, making it difficult to understand the necessity of this research.

W3. The experimental evaluation lacks crucial comparative analyses and does not fully explore the impact of parameter choices or dataset selection.


Detailed Comments

D1. Introduction and Motivation.
1) The introduction is difficult to follow and lacks clarity in explaining the motivation and significance of forest-based centrality measures over traditional ones.
2) The rationale for focusing on undirected graphs is weak, especially as undirected graphs can be seen as a subset of directed graphs. Clarifying why the study focuses specifically on undirected graphs would strengthen the justification.

D2. Definitions and Notation.
1) Terminology is introduced without adequate explanation, especially in distinguishing key terms like omega_uv versus omega_uu, which leads to confusion.
2) Notational inconsistencies, such as switching between similar symbols for related concepts, detract from clarity and make the paper harder to follow.

D3. Variance Reduction Technique.
1) The integration of isomorphic forest groups and neighbor-based estimators lacks a clear motivation as a unique contribution. A stronger explanation of its innovative value over existing variance reduction techniques is needed.
2) Including examples or diagrams (for instance, in Section 3.3) would improve comprehension of the estimators and their intended impact.

D4. Experimental Setup.
1) Comparative results with non-forest-based centrality measures are missing, which would provide a stronger context for understanding the empirical findings.
2) Details on parameter choices, particularly epsilon and delta in Algorithms 1 and 2, are lacking, leaving their effect on performance unexplored and the results less interpretable.

D5. Applicability and Contextual Fit.
1) The paper is aimed at data mining and graph analysis audiences but lacks a clear link to practical applications in these fields. A real-world case study demonstrating how these centrality measures could be applied to tasks in social networks would strengthen the paper's relevance

**Questions:**

Please refer to the detailed comments provided

**Reviewer Confidence:**

2: The reviewer is willing to defend the evaluation, but it is likely that the reviewer did not understand parts of the paper

**Scope:**

3: The work is somewhat relevant to the Web and to the track, and is of narrow interest to a sub-community

---

### Official Review · Reviewer_DrPW · 2024-12-01

**Novelty:** 5
**Technical Quality:** 6

**Review:**

__Summary__
This paper argues existing sampling based forest matrix estimation approaches mainly focus on directed graphs rather than widely existing undirected graphs, also the edge centrality measure computation is not efficient. To address these two issues this paper first combines the existing counting estimator with consideration of isomorphic spanning forests, with is then combined with the neighborhood based estimator idea, and proposes a node centrality estimator which is claimed to be unbiased and have lower variance. For edge centrality estimator, this paper proposes to also consider the contribution of different trees that an edge connects thus proposes another estimator, which is also claimed bo the unbiased and with lower variance. The algorithm design  as well as theoretical proofs are provided. Empirical performance as well as efficiency evaluation are also reported.

__Strengths__
* The motivation and proposed estimators are sound.
* The empirical study provides convincing evidence to the main claims.
* The main paper is easy to follow.

__Suggestion__
* Although I found the paper is very educative, I'm still missing (if I read correctly) why the existing estimators on directed graphs are not applicable (even with small modifications) to the undirected graph. It would be great to elaborate this more in the paper.

__Note: as I am not familiar with forrest matrix estimation. I am not able to assess the novelty of the proposed estimators nor the significance of the empirical results.__

**Questions:**

Please see suggestion

**Reviewer Confidence:**

2: The reviewer is willing to defend the evaluation, but it is likely that the reviewer did not understand parts of the paper

**Scope:**

4: The work is relevant to the Web and to the track, and is of broad interest to the community

---

### Official Review · Reviewer_LYo8 · 2024-12-01

**Novelty:** 5
**Technical Quality:** 5

**Review:**

This paper addresses the problem of approximating forest node centrality and edge centrality measures in large undirected networks. Building on the established relationship between the forest matrix entries and spanning forests, the authors propose two sampling-based methods for estimating these measures, leveraging the probabilistic interpretation of the related variables. The authors provide formal analysis and demonstrate the guarantees of the proposed sampling algorithms.

The paper is well-written and easy to follow, with the experimental results effectively showcasing the performance of the proposed algorithms.

**Questions:**

1. In the experiments, the authors refer to the proposed algorithms as IFGN and IFG, but in Figure 3, the names are listed as IFGN and IFGN+. This appears to be an inconsistency between the code and the written description.

2. In lines 226-227, the statement "It has been proven that the upper bound of FEC for edge" is made without a reference. Could the authors provide the necessary citation?

3. The process by which IFGN is modified to create the IFGN-E baseline is unclear. Could the authors clarify how this modification is done?

4. While the authors demonstrate that the variance improves compared to existing methods, they do not provide the exact variance. It would be theoritically more sound if the authors included the specific variance values.

**Reviewer Confidence:**

2: The reviewer is willing to defend the evaluation, but it is likely that the reviewer did not understand parts of the paper

**Scope:**

4: The work is relevant to the Web and to the track, and is of broad interest to the community

---

### Official Review · Reviewer_ATLG · 2024-12-04

**Novelty:** 5
**Technical Quality:** 6

**Review:**

This paper proposes new algorithms for computing forest centrality, which are demonstrated to outperform the state-of-the-art ones.

The main strengths of the paper are as follows:

S1) The paper provides a significant advancement to the state of the art in forest-centrality computation, which makes the paper relevant in terms of novelty and impact.

S2) The proposed algorithms are well-designed and sound, and they come with good technical contribution.

S3) The proposed methods achieve good experimental results.

The weaknesses of the paper are as follows:

W1) Accuracy evaluation is carried out by mean relative error only. This metric is a bit simplistic if applied to centrality measures, as it is sensitive to outliers and does not consider the ranking induced by the centrality scores. It would be good to complement such a metric with something less sensitive to outliers (median?) and with some metric that evaluate the difference in the ranking induced by the centrality scores.

W2) The research carried out in the paper is not contextualized to the Web context, as it would be required by the venue at hand. An option to overcome this limitation could be to add some case study that is relevent for the Web context.

**Questions:**

Please comment about W1) and W2).

**Ethics Review Description:**

No ethical issues.

**Reviewer Confidence:**

3: The reviewer is confident but not certain that the evaluation is correct

**Scope:**

3: The work is somewhat relevant to the Web and to the track, and is of narrow interest to a sub-community